# Enforcing Orderedness in SAEs to Improve Feature Consistency

## Abstract

Sparse autoencoders (SAEs) have been widely used for interpretability of neural networks, but their learned features often vary across seeds and hyperparameter settings. We introduce Ordered Sparse Autoencoders (OSAE), which extend Matryoshka SAEs by (1) establishing a strict ordering of latent features and (2) deterministically using every feature dimension, avoiding the sampling-based approximations of prior nested SAE methods. Theoretically, we show that OSAEs resolve permutation non-identifiability in settings of sparse dictionary learning where solutions are unique (up to natural symmetries). Empirically on Gemma2-2B and Pythia-70M, we show that OSAEs can help improve consistency compared to Matryoshka baselines.

## 1 Introduction

Sparse autoencoders (SAEs) have become central to unsupervised representation learning. Enforcing sparsity in the latent space yields interpretable, often disentangled features, enabling progress in clustering, visualization, and scientific discovery (Vincent et al., 2010; Coates et al., 2011; Ng, 2011). Yet despite their success, SAEs suffer from a critical shortcoming: the set of features they learn can vary across random seeds, initialization schemes, and hyperparameter settings, leading to poor reproducibility and undermining any mechanistic interpretation of individual latent dimensions (Song et al., 2025; Fel et al., 2025). Several strategies have been proposed to mitigate this instability. These include regularization techniques such as orthonormality penalties (Lee et al., 2025), structured sparsity constraints like group or tree sparsity (Jenatton et al., 2010), and post-hoc alignment or averaging of learned dictionaries across runs (Ghorbani et al., 2020).

One way to reduce the size of each equivalence class of solutions is to enforce structural constraints into the loss function. In particular, Matryoshka SAEs (Bussmann et al., 2025) are introduced to resolve a notion of hierarchy in feature learning. Their work defines an ordering on features by thir level of abstraction: "comma" is a lower-level feature than "punctuation mark". Matryoshka SAEs sample a small number of dictionary sizes per batch, thereby capturing multiscale features and partially breaking permutation symmetry. Despite these advances, Matryoshka SAEs treat features within each sampled group as exchangeable, a limitation from sampling only a handful of dictionary sizes (e.g., up to 10 per batch).

In this work, we introduce *Ordered Sparse Autoencoders* (OSAE), which extends Matryoshka SAEs by enforcing a strict ordering of latent dimensions. Drawing on the concept of nested dropout—which imposes an explicit ordering by stochastically truncating latent codes (Rippel et al., 2014)—OSAE treats each non-zero feature as its own dictionary size.

Our key contributions are:

- We propose Ordered Sparse Autoencoders (OSAE), which enforce deterministic feature ordering.
- We present theoretical results for ordered feature recovery by nested dropout loss in a special case of overcomplete sparse dictionary learning.

- We demonstrate improvement in feature consistency when using OSAEs on Gemma2-2B and Pythia-70M.

## 2  Problem setup

### 2.1  Preliminaries

Throughout, we use the following notation:

- $X = [x_1, \ldots, x_N] \in \mathbb{R}^{d \times N}$: the data matrix whose columns $x_i \in \mathbb{R}^d$ are samples.
- $E : \mathbb{R}^d \to \mathbb{R}^K$: the encoder mapping each input $x_i$ to a code $z_i = E(x_i)$.
- $D \in \mathbb{R}^{d \times K}$: the decoder or dictionary matrix, whose columns $d_j \in \mathbb{R}^d$ are basis atoms.
- $Z = E(X) \in \mathbb{R}^{K \times N}$: the code matrix, whose columns are the encoded vectors $z_i$.

We will consider two settings:

- **(Under)complete $(K \leq d)$.** $D$ spans a $K$-dimensional subspace (the PCA case).
- **Overcomplete $(K > d)$.** $D$ is a dictionary of $K$ atoms for sparse coding.

Define for all $\ell = 1, \ldots, K$:

$$\Lambda_\ell = \begin{bmatrix} I_\ell & 0 \\ 0 & 0 \end{bmatrix} \in \{0, 1\}^{K \times K},$$

$$\mathrm{Top}_m(z_i)_j = \begin{cases} z_{i,j}, & \text{if } |z_{i,j}| \text{ in top } m, \\ 0, & \text{otherwise,} \end{cases}$$

extended column-wise to $\mathrm{Top}_m(Z)$.

### 2.2  Nested dropout in the (under)complete setting

Consider the (under)complete linear autoencoder with representation dimension $K \leq d$. The standard reconstruction loss

$$\mathcal{L}_{\mathrm{AE}}(D, E) = \|X - D\,Z\|_F^2$$

recovers the top-$K$ principal subspace but leaves $D$ defined only up to an invertible transformation Baldi and Hornik (1989); Bourlard and Kamp (1988); Plaut (2018). Rippel et al. (2014) introduce the nested dropout loss, which minimizes

$$\mathcal{L}_{\mathrm{ND}}(D, E) = \mathbb{E}_{\ell \sim p_{\mathrm{ND}}} \|X - D\,\Lambda_\ell\,Z\|_F^2,$$

where $p_{\mathrm{ND}}(\ell)$ is a distribution over $\{1, \ldots, k\}$ with full support. With $D^\top D = I$, they theoretically show that this loss uniquely recovers the PCA eigenbasis in descending-eigenvalue order, rather than merely its subspace.

In the next section we extend this idea to the overcomplete, hard-$m$ sparse setting by inserting a Top-$m$ mask into the same expectation to obtain our Ordered Sparse Autoencoder (O-SAE).

### 2.3  Sparse dictionary learning

To understand the non-identifiability challenges faced by sparse autoencoders in the overcomplete regime, we first discuss classical sparse dictionary learning. The goal is to generalize PCA's fixed-size eigenbasis to an overcomplete dictionary of atoms that admits sparse representations. Concretely, each data vector $x_i$ is modeled as

$$X = [x_1, \ldots, x_N] \in \mathbb{R}^{d \times N}, \quad X = D\,Y,$$

where

$$D = [d_1, \ldots, d_K] \in \mathbb{R}^{d \times K}, \quad Y = [y_1, \ldots, y_N] \in \mathbb{R}^{K \times N},$$

with unit-norm atoms $\|d_j\|_2 = 1$ and sparse codes $\|y_i\|_0 \leq m \ll K$. That is, each sample $x_i$ is assumed to be generated by a linear combination of a small subset of the dictionary atoms.

Whereas PCA solves $\min \|X - DZ\|_F^2$ under a rank constraint $K \leq d$, sparse dictionary learning (SDL) tackles

$$\min_{D,Y} \|X - D\,Y\|_F^2 \quad \text{subject to} \quad \|y_i\|_0 \leq m,$$

an NP-hard problem due to the combinatorial nature of the $\ell_0$ sparsity constraint. In practice, this objective is typically approximated using greedy methods like orthogonal matching pursuit (OMP) Pati et al. (1993); Tropp and Gilbert (2007), alternating minimization algorithms such as K-SVD Aharon et al. (2006), or online optimization techniques Mairal et al. (2010).

A key challenge in SDL is the issue of non-identifiability: many dictionaries $D$ and code matrices $Y$ can produce the same reconstruction $X$, especially in the overcomplete setting. Even in the ideal noiseless case, identifiability of the ground-truth dictionary $D^*$ is only possible under strong structural assumptions.

**Spark and uniqueness.** The *spark* of a dictionary $D$,

$$\text{spark}(D) = \min\{\|z\|_0 : Dz = 0, \ z \neq 0\},$$

measures the size of the smallest linearly dependent set of atoms. If $\text{spark}(D) > 2m$, then any $m$-sparse representation $y = Dz$ is unique, guaranteeing identifiability in sparse coding. However, computing spark is NP-hard, so practitioners often rely on relaxed surrogate conditions:

- *Mutual coherence* $\mu(D) = \max_{i \neq j} |d_i^\top d_j|$, with $\mu(D)(m-1) < 1$ ensuring uniqueness via greedy methods such as OMP Tropp (2004).
- *Restricted isometry property* (RIP), which ensures $D$ approximately preserves the norms of all $m$-sparse vectors Candes and Tao (2005).

Recent work has begun applying these identifiability conditions to sparse autoencoders. In particular, Song et al. (2025) show that if a Top-$k$ SAE achieves exact sparsity and zero reconstruction error, then the encoder-decoder pair satisfies a round-trip condition that implies $\text{spark}(D) > 2k$, guaranteeing uniqueness of the learned features up to permutation and scaling. Our work builds on this by explicitly reducing *permutation ambiguity during training* itself.

### 2.4 $\ell$-PREFIX RECONSTRUCTION OBJECTIVE (Top-$m$).

We define:
$$\mathcal{L}_\ell(D, E) = \big\|X - D\,\Lambda_\ell\,\text{Top}_m(Z)\big\|_F^2.$$
This objective minimizes reconstruction loss when we use the top-$k$ codes and then truncate to the first $\ell$ dimensions. When $\ell = K$, this becomes the standard full-code reconstruction loss $\big\|X - D\,\text{Top}_k(Z)\big\|_F^2$ that standard top-$m$ SAEs minimize.

### 2.5 MATRYOSHKA SAE OBJECTIVE (Top-$m$).

Matryoshka SAEs (Bussmann et al., 2025) partition the $K$ atoms into a small collection of nested "groups" of increasing size $M = \{\ell_1 < \cdots < \ell_L\}$. At each training step, one group $1{:}\ell$ is sampled with probability $p_{\text{MSAE}}(\ell)$, and only atoms $d_1, \ldots, d_\ell$ (and their corresponding code entries) are used for reconstruction:

$$\mathcal{L}_{\text{MSAE}}(D, E) = \mathbb{E}_{\ell \sim p_{\text{MSAE}}}\big[\mathcal{L}_\ell(D, E)\big]$$
$$= \sum_{\ell \in M} p_{\text{MSAE}}(\ell)\,\big\|X - D\,\Lambda_\ell\,\text{Top}_m(Z)\big\|_F^2.$$

By enforcing reconstruction over only a handful of group sizes (e.g. 5–10 per batch), Matryoshka SAE captures multiscale features while partially breaking permutation symmetry within each group.

## 2.6 Nested dropout objective (Top-$m$).

We extend nested dropout (Rippel et al., 2014) by treating each individual atom $d_j$ as its own "group," so that sampling a prefix $\ell$ means retaining exactly atoms 1 through $\ell$ and dropping the rest. Let $p_{\mathrm{ND}}(\ell)$ be a distribution over $\{1, \ldots, m\}$ with full support. The nested-dropout loss is

$$\mathcal{L}_{\mathrm{ND}}(D, E) = \mathbb{E}_{\ell \sim p_{\mathrm{ND}}}\big[\mathcal{L}_\ell(D, E)\big]$$
$$= \sum_{\ell=1}^{m} p_{\mathrm{ND}}(\ell) \left\| X - D\, \Lambda_\ell \, \mathrm{Top}_m(Z) \right\|_F^2.$$

By covering all prefixes in expectation, this objective enforces a strict ordering of features.

## 2.7 Consistency evaluation

To quantify how reproducibly SAEs recover the same features across seeds, we adopt the *stability* metric from Fel et al. (2025). Let $D, D' \in \mathbb{R}^{d \times K}$ be two learned decoder matrices with unit-norm columns. We define

$$\mathrm{Stab}(D, D') = \max_{P \in \mathcal{P}} \frac{1}{K} \mathrm{tr}\big(D^\top P\, D'\big),$$

where $\mathcal{P}$ is the set of all $K \times K$ permutation matrices. This computes the average cosine similarity between matched atoms after optimal re-indexing via the Hungarian algorithm. When we compare a learned dictionary $D$ against the ground-truth dictionary $D^\star$, stability also serves as a *feature recovery fidelity* metric, indicating how accurately the true atoms are recovered.

## 2.8 Orderedness evaluation

To quantify how similarly in order SAEs recover features across seeds, we introduce an orderedness metric. Let $D, D' \in \mathbb{R}^{d \times K}$ be two dictionaries, each with an inherent ordering of their atoms (e.g. by frequency, abstraction, or another criterion). After matching each atom $d_j$ in $D$ to its best-corresponding atom $d'_{\mu(j)}$ in $D'$ via the Hungarian algorithm, we obtain a permutation vector

$$\mu = \big(\mu(1), \mu(2), \ldots, \mu(K)\big) \in \{1, \ldots, K\}^K.$$

We then define the *orderedness* between $D$ and $D'$ as the Spearman rank correlation between their index sequences:

$$\mathrm{Ord}(D, D') = \mathrm{Spearman}\big((1, \ldots, K), \mu\big) = 1 - \frac{6 \sum_{j=1}^{K} \big(j - \mu(j)\big)^2}{K(K^2 - 1)}.$$

A value $\mathrm{Ord}(D, D') = 1$ indicates perfect matching of their orderings.

## 3 Exact recovery of ordered features

Define the domain of optimization to be

$$\mathcal{F} = \big\{(D, E) \mid D = [d_1, \ldots, d_K] \in \mathbb{R}^{d \times K}, \ \|d_j\|_2 = 1 \quad (\forall j = 1, \ldots, K), \ E : \mathbb{R}^d \to \mathbb{R}^K\big\}.$$

In other words, $\mathcal{F}$ consists of all decoder–encoder pairs $(D, E)$ in which each dictionary atom $d_j$ has unit $\ell_2$-norm, and $E$ is an arbitrary mapping from $\mathbb{R}^d$ to $\mathbb{R}^K$.

Suppose $X = D^* Y^*$, where $D^* = [d_1^*, \ldots, d_K^*] \in \mathbb{R}^{d \times K}$ has unit-norm columns satisfying $\mathrm{spark}(D^*) > 2m$, and $Y^* \in \mathbb{R}^{K \times N}$ $m$-sparse columns.

**Lemma 3.1.** *Any minimiser of $\mathcal{L}_{\mathrm{ND}}$ also minimises the full-prefix loss $\mathcal{L}_k$. That is,*

$$\mathrm{argmin}_{(D,E) \in \mathcal{F}} \mathcal{L}_{\mathrm{ND}} \ \subseteq \ \mathrm{argmin}_{(D,E) \in \mathcal{F}} \mathcal{L}_{\mathrm{K}}.$$

The proof is deferred to Appendix A

**Theorem 3.1.** *[Exact ordered recovery under spark condition] Assume the columns of $Y^*$ are nonnegative (to resolve sign ambiguity) and "true" atoms are ordered so that*

$$|\{\, i : y_{1,i}^* > 0 \,\}| \;\geq\; |\{\, i : y_{2,i}^* > 0 \,\}| \;\geq\; \cdots \;\geq\; |\{\, i : y_{K,i}^* > 0 \,\}|.$$

*Then any global minimiser $(\widehat{D}, \widehat{E}) \in \mathcal{F}$ of the nested-dropout loss $\mathcal{L}_{\mathrm{ND}}$ satisfies*

$$\widehat{D} = D^*, \qquad \widehat{E}(X) = Y^*$$

*Proof.* By Lemma 3.1, any minimiser of $\mathcal{L}_{\mathrm{ND}}$ also minimises the full-prefix loss $\mathcal{L}_K$. Hence $(\widehat{D}, \widehat{Y})$ with $\widehat{Y} = \mathrm{Top}_{\mathrm{m}}\big(\widehat{E}(X)\big)$ satisfies

$$\widehat{D}\,\widehat{Y} = X, \qquad \|\widehat{y}_i\|_0 \leq m.$$

The uniqueness result under the spark condition then gives

$$\widehat{D} = D^* P S, \quad \widehat{Y} = S^{-1} P^\top Y^*,$$

for some permutation matrix $P$ and invertible diagonal $S$.

Since all columns of $Y^*$ and $\widehat{Y}$ are nonnegative, $S$ must be the identity (no sign-flips or rescaling). Finally, because the atoms were assumed ordered by their sparsity-support frequencies, the only permutation that preserves that ordering is the identity. Hence $P = I$ and

$$\widehat{D} = D^*, \quad \widehat{Y} = Y^*,$$

as claimed. □

## 3.1 Toy Gaussian model

Theorem 3.1 gives theoretical guarantees that SAEs minimising nested dropout loss can achieve perfect consistency and orderedness under certain conditions of the data and sparsity. We evaluate under the following synthetic generative model:

1. **Parameters.** Fix dimensions $d, n, K$ and sparsity level $m \leq d$. Assume an *ordering distribution* $\pi = (\pi_1, \ldots, \pi_K)$ with $\pi_1 \geq \pi_2 \geq \cdots \geq \pi_K > 0$ and $\sum_{j=1}^K \pi_j = 1$.

2. **Dictionary generation.**

$$D = [\, d_1, \ldots, d_K \,] \in \mathbb{R}^{d \times K}, \qquad d_j \overset{\mathrm{iid}}{\sim} \mathcal{N}\big(0, \tfrac{1}{d} I_d\big), \quad d_j \leftarrow \frac{d_j}{\|d_j\|_2}.$$

3. **Code generation.** For each sample $i = 1, \ldots, n$:

   (a) Sample a support of size $m$ by drawing indices without replacement according to $\pi$:
   $$S_i \;\sim\; \mathrm{MultisetSample}(\pi, m).$$

   (b) Let
   $$y_i \in \mathbb{R}^K, \quad (y_i)_j = \begin{cases} z_{ij}, & j \in S_i, \\ 0, & j \notin S_i, \end{cases} \quad z_{ij} \overset{\mathrm{iid}}{\sim} \mathcal{N}(0,1).$$

   (c) For the purposes of this toy model, we remove sign ambiguity: $y_i \leftarrow |y_i|$.

   Collect into $Y = [\, y_1, \ldots, y_n \,] \in \mathbb{R}^{K \times n}$.

4. **Data matrix.**
$$X = D\,Y \;\in\; \mathbb{R}^{d \times n}.$$

Under this model, atoms with smaller index $j$ appear more frequently in the data (higher $\pi_j$), inducing a ground-truth ordering that we will attempt to recover via O-SAEs. Since the dictionary is drawn from a standard Gaussian ensemble, the spark condition for uniqueness is satisfied with high probability (Hillar and Sommer, 2015).

**Unit sweeping.**  Nested dropout samples a truncation index $b \sim p_B(\cdot)$, so gradients onto late units shrink exponentially with index. To avoid starving these units, we employ *unit sweeping* (Rippel et al., 2014): once a lower-index unit has effectively converged, we *freeze* its encoder row and decoder column (stop backprop through that unit) and continue training the remaining, unfrozen units. Practically, we use a simple "clockwork" schedule that freezes one additional unit every $T$ epochs (from $1 \to K$), after a short burn-in; frozen units remain in the forward pass, and we renormalize decoder columns to unit norm after each freeze. Results in Fig. 4 and Table 1 use unit sweeping, which we find improves stability and ordered recovery in this setting.

**Evaluation.**  We evaluate Ordered SAE, Matryoshka SAE (Fixed and Random, with five groups), and Vanilla top-$m$ SAEs on the toy model above with $(d, K, m, N) = (80, 100, 5, 100,000)$ and a Zipf support prior $\pi_j \propto j^{-\alpha}$, $\alpha = 1.2$, which induces a strict ground-truth ordering. Hyperparameters are selected by lowest validation reconstruction error at the target sparsity. We use the warmup schedule for $k$ (from $K$ down to $m$) and train with *unit sweeping*. Qualitative recovery patterns are shown in Fig. 1 for Fixed MSAE and OSAE; full results can be found in Fig. 4. Quantitative results (mean $\pm$ std) are summarized in Table 1. Importantly, $\mathrm{Stab}(D, D')$ is averaged over $\binom{10}{2} = 45$ seed pairs, while $\mathrm{Stab}(D, D^*)$ and $\mathrm{Ord}(D, D^*)$ are averaged over 10 seed $\to D^*$ comparisons; reconstruction loss is MSE on a held-out set.

For each model, we choose the hyperparameter configuration that achieves the lowest validation reconstruction error at the target sparsity level.

To stabilize training and improve recovery, we adopt a warmup strategy for top-$k$ truncation. Specifically, we begin training with a large truncation size $k_{\mathrm{init}} \geq m$ (typically $k_{\mathrm{init}} = K$) and gradually decrease $k$ to the target sparsity $m$ over a fixed number of epochs. This schedule smooths the optimization landscape by initially allowing dense activations before progressively enforcing sparsity. We find this warmup significantly improves convergence, particularly for O-SAE and Matryoshka models where early features receive higher training pressure. Additional results are shown in Appendix B.

| Model | $\mathrm{Stab}(D,D')$ | $\mathrm{Stab}(D,D^*)$ | $\mathrm{Ord}(D,D^*)$ | Reconstruction loss |
|---|---|---|---|---|
| Vanilla SAE | 0.572 (0.00964) | 0.479 (0.0104) | 0.0162 (0.128) | 0.0257 (0.000841) |
| Fixed MSAE | 0.538 (0.0114) | 0.502 (0.0160) | 0.119 (0.673) | 0.0339 (0.00635) |
| Random MSAE | 0.531 (0.0150) | 0.480 (0.0106) | 0.0544 (0.0760) | 0.0309 (0.00366) |
| **OSAE (ours)** | **0.664 (0.0191)** | **0.814 (0.0195)** | **0.734 (0.0758)** | **0.00725 (0.000746)** |

Table 1: Summary metrics on the toy Gaussian model (mean $\pm$ std).

We found it surprising that our O-SAEs achieved the lowest reconstruction loss here despite possessing inductive biases that restrict the solution class. Thus for a controlled evalation with minimal influence from hyperparameter bias, we ran further evaluations for O-SAEs in Appendix B.3 directly extending Song et al. (2025)'s synthetic experiments. There, O-SAEs achieve a similar consistency and higher orderedness compared to baseline architectures despite a higher MSE loss, helping support our findings here.

## 4  RESULTS ON EMPIRICAL DATA

### 4.1  TEXT-BASED EVALUATIONS ON GEMMA-2 2B

**Setup.**  We evaluate the orderedness and stability of pairs of random seeds for Ordered SAEs and Matryoshka SAE variants trained on Gemma-2 2B (Team et al., 2024). We train these SAEs with dictionary size 4096 on layer 12 with K=80, collecting activations from Gemma-2 2B on an uncopyrighted portion of the Pile (Gao et al., 2020). We use the same piecewise distribution for the O-SAE as Matryoshka baselines for these experiments. Unit sweeping is not employed for these experiments. We note that O-SAEs have slightly worse reconstruction loss compared to baselines, potentially due to a restricted solution

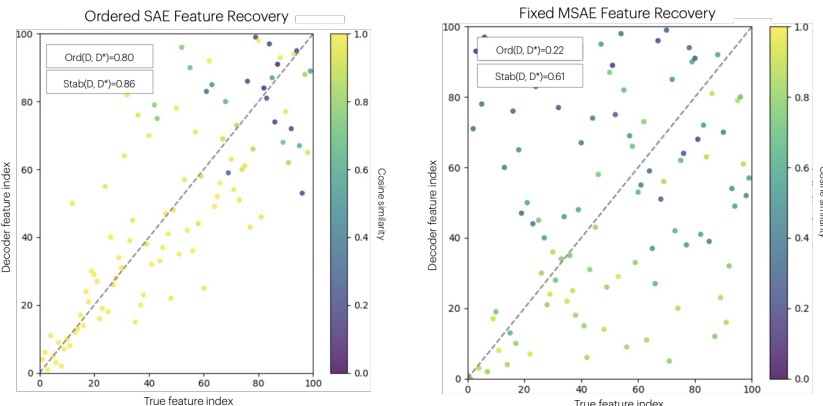

Figure 1: Recovery across SAE variants on a Gaussian toy model with $(d, K, m, N) = (80, 100, 5, 100\,000)$. Each panel plots the Hungarian matching between learned decoder atoms $D$ and ground truth $D^*$ (one dot per matched pair; color encodes cosine similarity). Ordered SAEs achieve higher stability $\mathrm{Stab}(D, D^*)$ (mean matched cosine) and higher orderedness $\mathrm{Ord}(D, D^*)$ (order agreement), meaning they recover features more faithfully and in order.

space or, for a simpler explanation, our limited hyperparameter sweeps. Thus, we compare checkpoints with similar loss values for a fair orderedness and stability comparison.

**O-SAEs achieve greater orderedness and stability in Gemma-2 2B.** In Figure 2, we demonstrate that O-SAEs achieve better orderedness (defined in 2.8) than the Fixed-prefix MSAE and Random-prefix MSAE with 5 groups. Although the overall average stability (defined in 2.7) is lower for O-SAE than the Matryoshka variants, we find that the most significant features have higher stability. For both metrics, we calculate the truncated metrics for the first $p$ prefix length features. We see that O-SAEs have a lower prefix-length stability compared to the Random MSAE at a crossover point between 1024 and 2048 features, where features are less significant in the feature ordering.

### 4.2 Consistency across datasets on Pythia 70m

We furthermore test the generalizability of orderedness and stability of O-SAEs trained on a different model, Pythia-70M (Biderman et al., 2023), and also evaluate consistency when activations are collected on different datasets. Both the Pile (Gao et al., 2020) and Dolma (Soldaini et al., 2024) are diverse and general pretraining mixes, so we select these datasets to represent a wide distribution of language while not drawing from the exact same data sources. If SAEs learn a good representation of general language, we ideally hope that the representations are consistent across datasets. We evaluate (1) same-dataset consistency similar to the previous section and (2) cross-dataset consistency by evaluating pairs of SAEs where one is trained on activations on the Pile and the other on activations from Dolma.

**Setup.** We train with the same hyperparameters as Gemma2-2B, although Pythia-70M is much smaller and has fewer layers. We use layer 3 of Pythia-70M, roughly halfway, to approximate the same position as layer 12 in Gemma2-2B. We train five seeds per (model, dataset) pair. In these experiments, we evaluate checkpoints after training for 50M tokens, reaching 0.02-0.03 recon loss.

**Similar trends on the Pile and Dolma datasets.** In Figure 3a, we evaluate same-dataset orderedness and stability on the SAEs trained on the Pile and Dolma. We see similar trends of higher orderedness for O-SAEs while stability degrades after time. All SAE variants maintain similar performance between the Pile and Dolma in same-dataset evaluations.

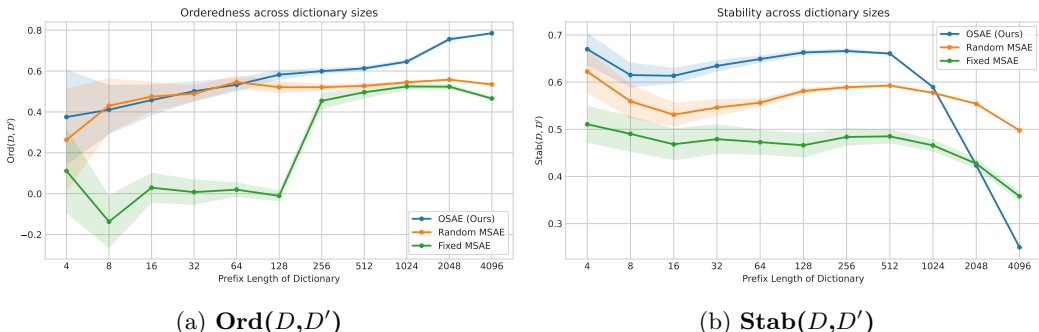

(a) **Ord**$(D,D')$            (b) **Stab**$(D,D')$

Figure 2: SAEs trained on Gemma2-2B. (a) Orderedness evaluated at different prefix lengths. O-SAE's have the most consistently ordered features almost reaching an average **Ord**$(D,D')$ of 0.8. As expected, we observe orderedness close to 0.0 for the first 128 features of the Fixed MSAE since the first group size is 128, whereafter it jumps up to values between around 0.5. (b) O-SAEs have high stability for the first portion of features, before a sharp decline for later features. $\binom{9}{2} = 36$ pairs of seeds are evaluated per method and 95% confidence intervals are visualized.

We include additional visualizations of how measures of orderedness and stability evolve with increasing amounts of training on same-dataset evaluations on the Pile and Dolma in Figure 11 and 12 in Appendix C. Interestingly, there are some cases for both O-SAEs and Random MSAEs where as training progresses, high levels of orderedness and stability in earlier prefixes decrease while these measures at later prefixes increase. We suspect this may be an artifact of the probability distribution or over-training, but we plan to run more ablations.

**O-SAEs also improve orderedness in cross-dataset comparisons.** In Figure 3b, we evaluate orderedness and stability of SAEs in cross-dataset settings, where one SAE is trained on the Pile and the other on Dolma. We observe that cross-dataset orderedness and stability matches trends from same-dataset results in Figure 3a, with a decline of around 0.1-0.2 from same-dataset results.

When we use same-seed initialization, O-SAEs achieve near-1.0 value in Orderedness and stability of 0.8 at full prefix length, when trained on different datasets. Orderedness and stability also increase in early prefix dictionary positions compared to the cross-seed settings for both O-SAEs and Random MSAEs; however, increases are more substantial for O-SAEs. We hypothesize the greater jump in full-prefix orderedness and stability metrics for O-SAEs are because they do not update their latter indices as much as earlier indices; this is further supported by Figure 13 and 14 in Appendix C showing minimal full-prefix changes in orderedness when comparing trained models against their initialization state for O-SAEs and significant full-prefix drops for Random MSAEs when comparing trained models against their initialization state.

## 5 LIMITATIONS

Our theoretical guarantees assume idealized conditions—sparsity and uniqueness assumptions and a well-specified ordering prior—that may not hold exactly in real data. If the imposed order is misspecified, the objective can over-regularize and suppress equally valid alternative bases. The training cost for O-SAEs is higher because covering prefixes requires more compute, so practical deployments may require approximations or careful schedule design. Performance is also sensitive to design choices such as the prefix distribution and unit sweeping. Finally, our empirical study focuses on controlled settings intended to evaluate the mechanism rather than to exhaustively benchmark task performance across architectures.

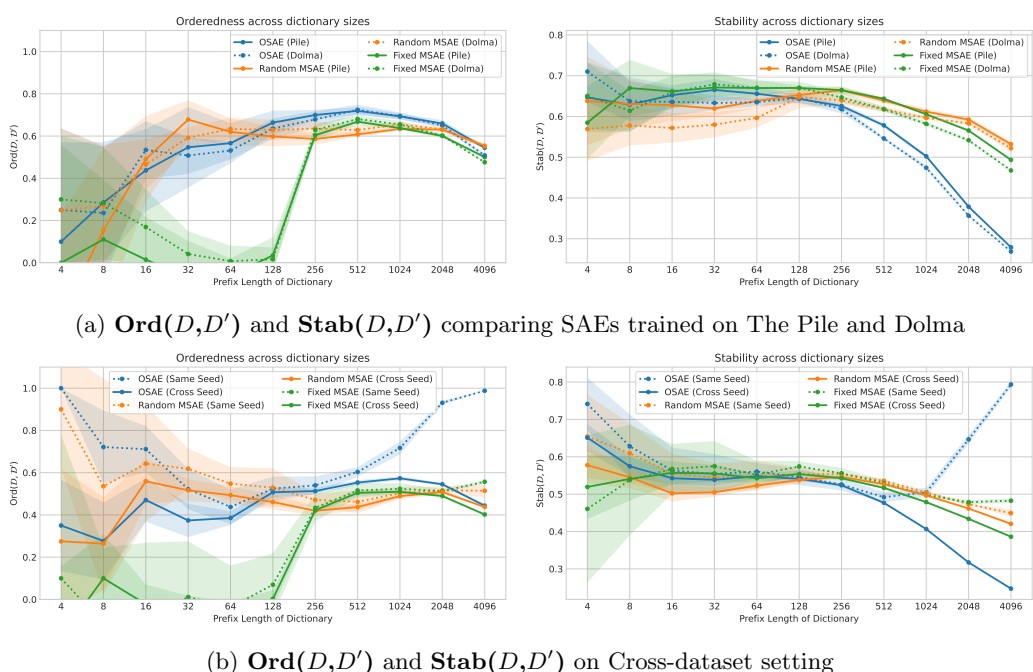

(a) $\mathbf{Ord}(D,D')$ and $\mathbf{Stab}(D,D')$ comparing SAEs trained on The Pile and Dolma

(b) $\mathbf{Ord}(D,D')$ and $\mathbf{Stab}(D,D')$ on Cross-dataset setting

Figure 3: SAEs trained on Pythia-70M (a) O-SAEs demonstrate an improvement in ordered-ness over Random MSAEs and Fixed MSAEs on the Pile and Dolma after prefix length 128. O-SAE stability is likewise stronger than Random MSAE on both datasets for the beginning features, but crosses over at around prefix length 128. Fixed MSAE stability is higher than O-SAE, but has lower orderedness. (n=10). (b) In the cross-dataset, cross-seed setting we observe that O-SAE has modest improvements in orderedness against Random MSAE and sizable stability gains against Random MSAE before prefix 128. O-SAEs and Random MSAEs demonstrate improvements to orderedness and stability when using the same seed despite training on different datasets; however, the improvements are larger for O-SAEs. (Cross-Seed: n=20. Same-Seed: n=5)

## 6  DISCUSSION

By optimizing an ordered, nested-prefix objective, we *shrink the solution class* of sparse au-toencoders. The plain reconstruction loss admits large equivalence classes (permutations and near-mixings of features) that undermine reproducibility; the ordered objective effectively selects a canonical basis. This narrowing of admissible solutions is a training-time struc-tural prior, which (i) constrains the hypothesis space, (ii) alters the optimization geometry toward a feature-curriculum, and (iii) makes the learned representation more comparable across runs and hyperparameters. We view ordering as a general mechanism for enforc-ing identifiability into overcomplete models, complementary to sparsity and incoherence assumptions in classical dictionary learning. Furthermore, we provide empirical results for O-SAEs on Gemma2-2B and Pythia 70m, trained on the Pile and Dolma, that demonstrate greater orderedness and stability in earlier features, while sometimes at the cost of lower stability in later, less-significant features.

## Use of LLMs

We made limited use of LLMs during paper preparation. Specifically, we used them to help write scripts for generating plots, and to suggest edits aimed at improving clarity of the text. All scientific claims are validated by the authors.

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

## A  Proofs

*Proof of Lemma 3.1.* Assume for contradiction that $(D^\star, E^\star) \in \mathcal{F}$ globally minimizes $\mathcal{L}_{\mathrm{ND}}$ but *does not* minimize $\mathcal{L}_K$. Write

$$Z^\star = E^\star(X), \qquad R^\star = X - D^\star \operatorname{Top}_m(Z^\star), \qquad \Delta = \frac{1}{N}\|R^\star\|_F^2 = \mathcal{L}_K(D^\star, E^\star) > 0.$$

Denote the $K$th row of $\operatorname{Top}_m(Z^\star)$ by $y_{K\cdot}^\star \in \mathbb{R}^n$.

If $y_{K\cdot}^\star = 0$ then none of the prefix losses ever see atom $K$, so we could remove it entirely and re-index, strictly reducing the nested-dropout loss unless $R^* = 0$. Hence for a strict counterexample we may assume $y_{K\cdot}^\star \neq 0$."

Set

$$v = R^\star y_{K\cdot}^\star = \sum_{i=1}^n r_{\cdot i}^\star y_{K,i}^\star \in \mathbb{R}^d,$$

which is nonzero since $R^\star \neq 0$ and $y_{K\cdot}^\star \neq 0$. Define

$$u = \frac{(I - d_K^\star (d_K^\star)^\top) v}{\|(I - d_K^\star (d_K^\star)^\top) v\|_2}.$$

Then $u$ is unit-length, $u \perp d_K^\star$, and

$$\sum_{i=1}^n y_{K,i}^\star (u^\top r_{\cdot i}^\star) = u^\top \Big(\sum_i r_{\cdot i}^\star y_{K,i}^\star\Big) = u^\top v = \big\|(I - d_K^\star (d_K^\star)^\top) v\big\|_2 > 0.$$

Now for small $\epsilon > 0$ define a perturbation of only the $K$th atom:

$$d_K^{\mathrm{new}} = \frac{d_K^\star + \epsilon u}{\|d_K^\star + \epsilon u\|_2} = d_K^\star + \epsilon u - \tfrac{1}{2}\epsilon^2 d_K^\star + O(\epsilon^3),$$

and leave $E^\star$ (hence $\operatorname{Top}_m(Z)$) unchanged. All other atoms remain as in $D^\star$, so $(D^\epsilon, E^\star) \in \mathcal{F}$.

Note that every prefix $\ell < K$ satisfies

$$D^\epsilon \Lambda_\ell \operatorname{Top}_m(Z^\star) = D^\star \Lambda_\ell \operatorname{Top}_m(Z^\star),$$

hence $\mathcal{L}_\ell(D^\epsilon, E^\star) = \mathcal{L}_\ell(D^\star, E^\star)$ for all $\ell < K$. Therefore

$$\mathcal{L}_{\mathrm{ND}}(D^\epsilon, E^\star) = \sum_{\ell=1}^K p_\ell \, \mathcal{L}_\ell(D^\epsilon, E^\star) = p_K \, \mathcal{L}_K(D^\epsilon, E^\star) + \sum_{\ell < K} p_\ell \, \mathcal{L}_\ell(D^\star, E^\star).$$

It suffices to show $\mathcal{L}_K(D^\epsilon, E^\star) < \mathcal{L}_K(D^\star, E^\star)$.

Since $\operatorname{Top}_m(Z)$ is unchanged,

$$R^\epsilon = X - D^\epsilon \operatorname{Top}_m(Z^\star) = R^\star - \big(d_K^{\mathrm{new}} - d_K^\star\big) y_{K\cdot}^{\star\,\top}.$$

Using $d_K^{\mathrm{new}} - d_K^\star = \epsilon u - \tfrac{1}{2}\epsilon^2 d_K^\star + O(\epsilon^3)$, one finds, entrywise,

$$r_{\alpha i}^\epsilon = r_{\alpha i}^\star - \epsilon u_\alpha y_{K,i}^\star + O(\epsilon^2).$$

Squaring and summing,

$$\|R^\epsilon\|_F^2 = \sum_{\alpha,i} (r_{\alpha i}^\epsilon)^2 = \sum_{\alpha,i} (r_{\alpha i}^\star)^2 - 2\epsilon \sum_i y_{K,i}^\star (u^\top r_{\cdot i}^\star) + O(\epsilon^2).$$

By our choice of $u$, the coefficient $\sum_i y_{K,i}^\star (u^\top r_{\cdot i}^\star)$ is strictly positive, so for sufficiently small $\epsilon$ the linear term makes $\|R^\epsilon\|_F^2 < \|R^\star\|_F^2$. Equivalently,

$$\mathcal{L}_K(D^\epsilon, E^\star) = \tfrac{1}{N}\|R^\epsilon\|_F^2 < \tfrac{1}{N}\|R^\star\|_F^2 = \mathcal{L}_K(D^\star, E^\star).$$

Since all $\mathcal{L}_{\ell < K}$ remain fixed,

$$\mathcal{L}_{\mathrm{ND}}(D^\epsilon, E^\star) < \mathcal{L}_{\mathrm{ND}}(D^\star, E^\star),$$

contradicting the global minimality of $(D^\star, E^\star)$. Therefore no residual can remain, and every minimiser of $\mathcal{L}_{\mathrm{ND}}$ must satisfy $\|R\|_F = 0$, i.e. also minimise $\mathcal{L}_K$. $\qquad\square$

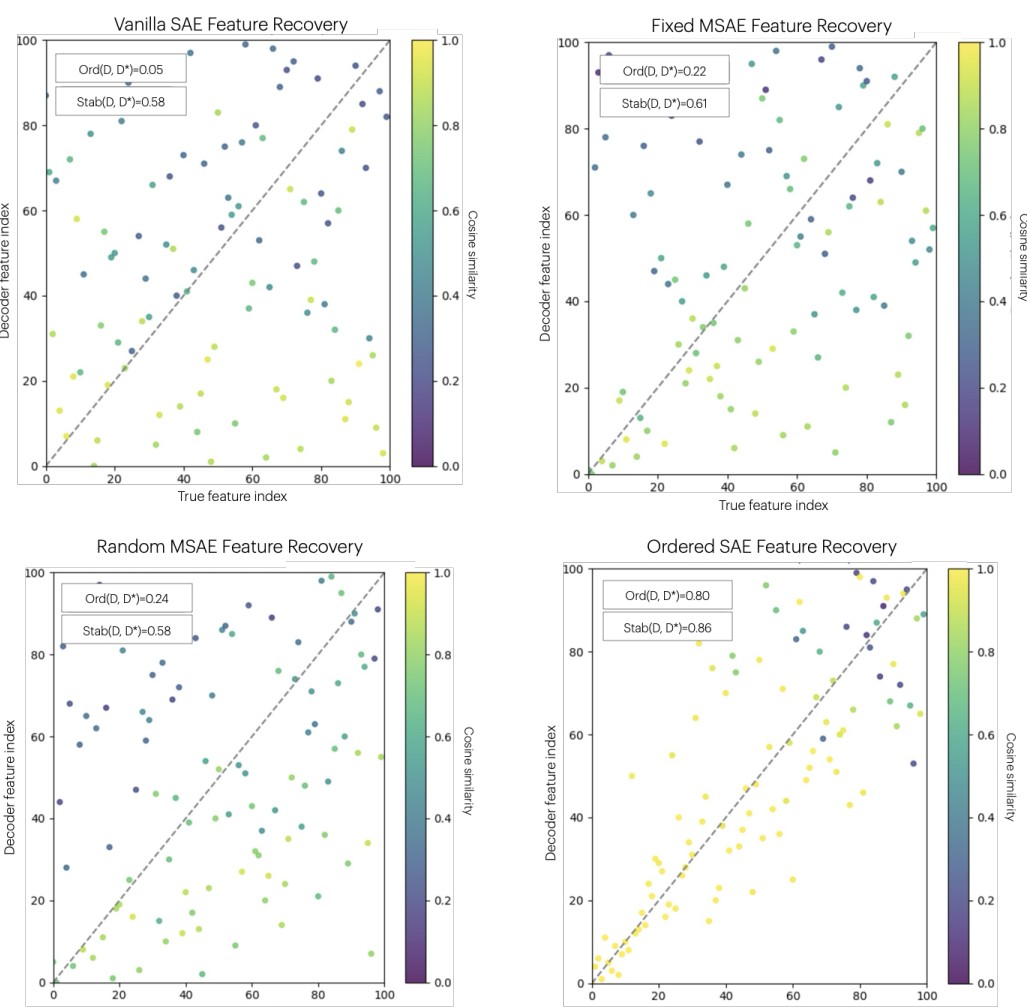

Figure 4: Recovery across SAE variants on a Gaussian toy model with $(d, K, m, N) = (80, 100, 5, 100\,000)$. Each panel plots the Hungarian matching between learned decoder atoms $D$ and ground truth $D^*$ (one dot per matched pair; color encodes cosine similarity). Ordered SAEs achieve higher stability $\mathrm{Stab}(D, D^*)$ (mean matched cosine) and higher orderedness $\mathrm{Ord}(D, D^*)$ (order agreement), meaning they recover features more faithfully and in order.

## B  TOY MODEL RESULTS

### B.1  ACTIVATION–STREAM STABILITY AND ORDEREDNESS

Let $Z^{(s)} = E^{(s)}(X) \in \mathbb{R}^{K \times N}$ be the code matrix for seed $s$ (rows are unit activations over the evaluation set), and let $Y^* \in \mathbb{R}^{K \times N}$ be the ground-truth activations. We replace decoder-space cosine with *Pearson correlation* on the activation stream and compute stability via Hungarian assignment:

$$R_{ij}^{(\rho)} = \mathrm{corr}\big(Z_{i:}^{(s)}, Y_{j:}^*\big), \qquad S_{ij}^{(\rho)} = \mathrm{corr}\big(Z_{i:}^{(a)}, Z_{j:}^{(b)}\big).$$

With $P$ the optimal permutation matrix, we report

$$\mathrm{Stab}_Z(Z^{(s)}, Y^*) = \tfrac{1}{K}\,\mathrm{tr}\big(R^{(\rho)}P\big), \qquad \mathrm{Stab}_Z(Z^{(a)}, Z^{(b)}) = \tfrac{1}{K}\,\mathrm{tr}\big(S^{(\rho)}P\big),$$

and define orderedness on the induced permutation as

$$\mathrm{Ord}_Z = \mathrm{Spearman}\big((1, \ldots, K), (\mu(1), \ldots, \mu(K))\big), \quad \text{where } \mu \text{ is read off from } P.$$

These $Z$-based measures complement the decoder–cosine results in the main text. Whereas Fig. 4 visualizes only matched pairs, the figures below show the *full $K \times K$ similarity fields*. In the top row we also show a small raster (50 evaluation inputs) to compare activation patterns $Y^*$ vs. $Z^{(0)}$ vs. $Z^{(1)}$. *Note:* for Vanilla and Matryoshka models, activations can be much larger than $Y^*$, so per-panel normalization makes the rasters not directly comparable in scale; this effect is less pronounced for the ordered model.

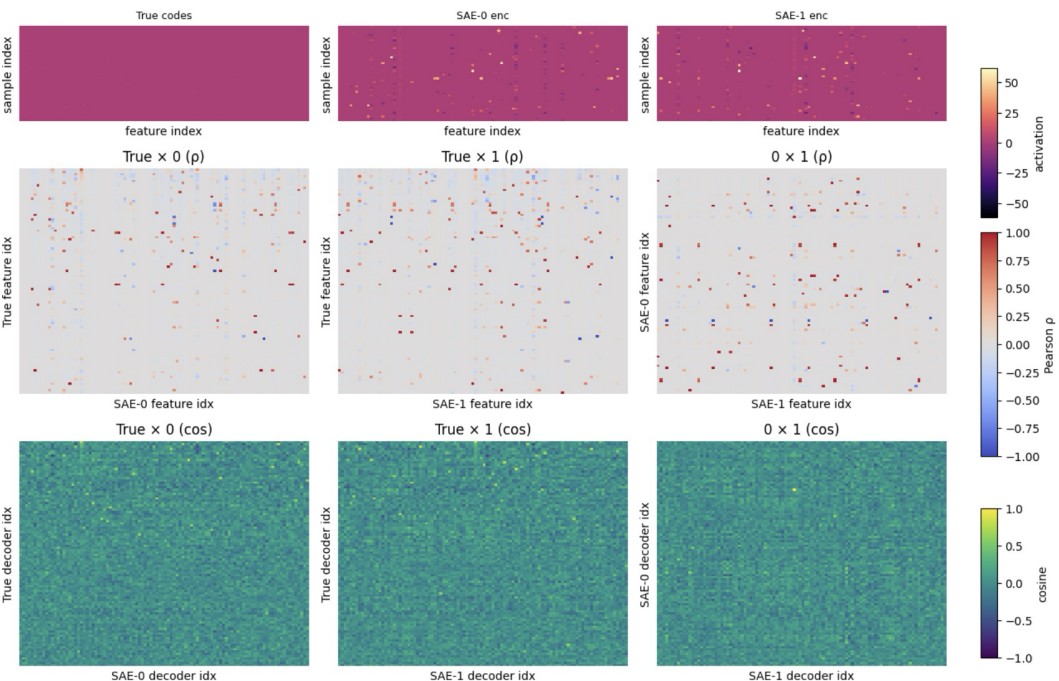

Figure 5: **Vanilla SAE (example seed pair).** *Top:* activation rasters for 50 eval inputs (left: $Y^*$, middle: $Z^{(0)}$, right: $Z^{(1)}$). *Middle:* all-pairs activation–Pearson matrices (left: $Z^{(0)}$ vs. $Y^*$, middle: $Z^{(1)}$ vs. $Y^*$, right: $Z^{(0)}$ vs. $Z^{(1)}$) used for $\mathrm{Stab}_Z$ and $\mathrm{Ord}_Z$. *Bottom:* all-pairs decoder–cosine matrices (left: $D^{(0)}$ vs. $D^*$, middle: $D^{(1)}$ vs. $D^*$, right: $D^{(0)}$ vs. $D^{(1)}$); this extends Fig. 4 from matched pairs to all pairs.

### B.2 Additional dictionary sizes

We repeat the activation–stream analysis at smaller widths with matching sparsities, $(K, m) \in \{(10, 2), (30, 3), (50, 5)\}$. Each panel uses the same three-row layout as Appendix B.1: *top*—activation rasters for 50 evaluation inputs (left: $Y^*$, middle: $Z^{(0)}$, right: $Z^{(1)}$); *middle*—all-pairs activation–Pearson matrices (left: $Z^{(0)}$ vs. $Y^*$, middle: $Z^{(1)}$ vs. $Y^*$, right: $Z^{(0)}$ vs. $Z^{(1)}$) used for $\mathrm{Stab}_Z$ and $\mathrm{Ord}_Z$; *bottom*—all-pairs decoder–cosine matrices (left: $D^{(0)}$ vs. $D^*$, middle: $D^{(1)}$ vs. $D^*$, right: $D^{(0)}$ vs. $D^{(1)}$). Figures show a *single example seed pair* for brevity; Matryoshka variants are omitted. Note that in Vanilla, activations can be much larger than $Y^*$, so per-panel normalization of the top row can make scales not directly comparable; this effect is less pronounced for O-SAE.

### B.3 Zipfian toy model: high consistency with moderate orderedness

**Setup.** We evaluate Ordered SAEs (O-SAEs; "Ordered TopK" in the legend) on a synthetic Zipfian activation process. Following Song et al. (2025), inputs live in $\mathbb{R}^{16}$ with an overcomplete ground-truth dictionary of 32 atoms; $k = 3$ features are active per sample, and

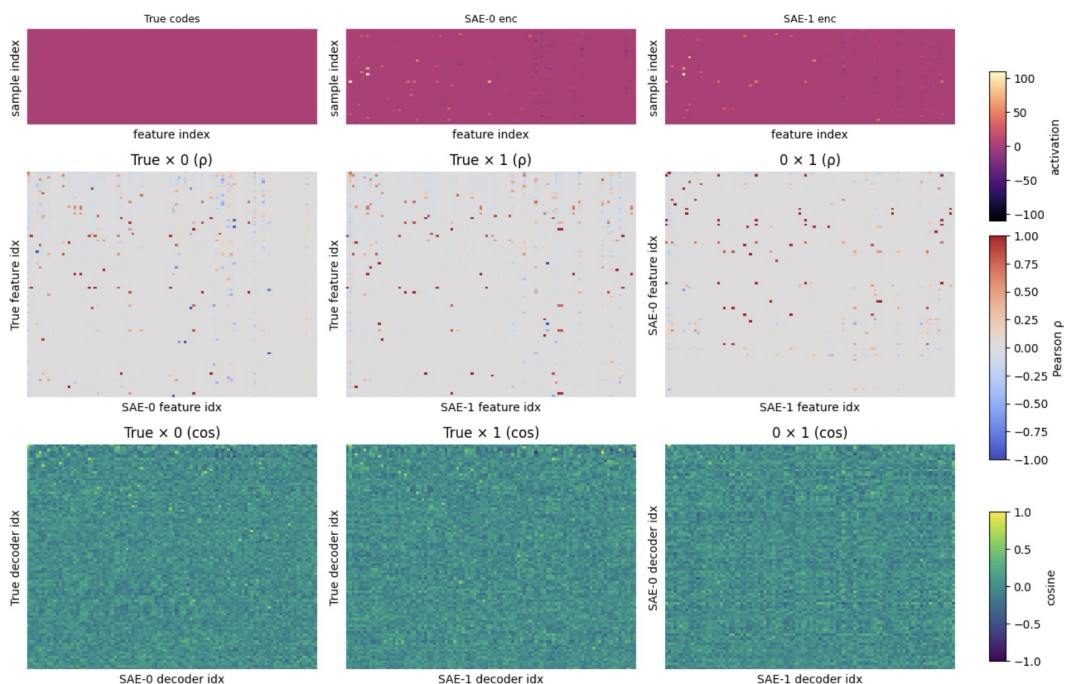

Figure 6: **Fixed MSAE (example seed pair).** Same layout as Fig. 5: top—activation rasters ($Y^*$, $Z^{(0)}$, $Z^{(1)}$); middle—all-pairs activation–Pearson ($Z^{(0)}$ vs. $Y^*$, $Z^{(1)}$ vs. $Y^*$, $Z^{(0)}$ vs. $Z^{(1)}$); bottom—all-pairs decoder–cosine ($D^{(0)}$ vs. $D^*$, $D^{(1)}$ vs. $D^*$, $D^{(0)}$ vs. $D^{(1)}$).

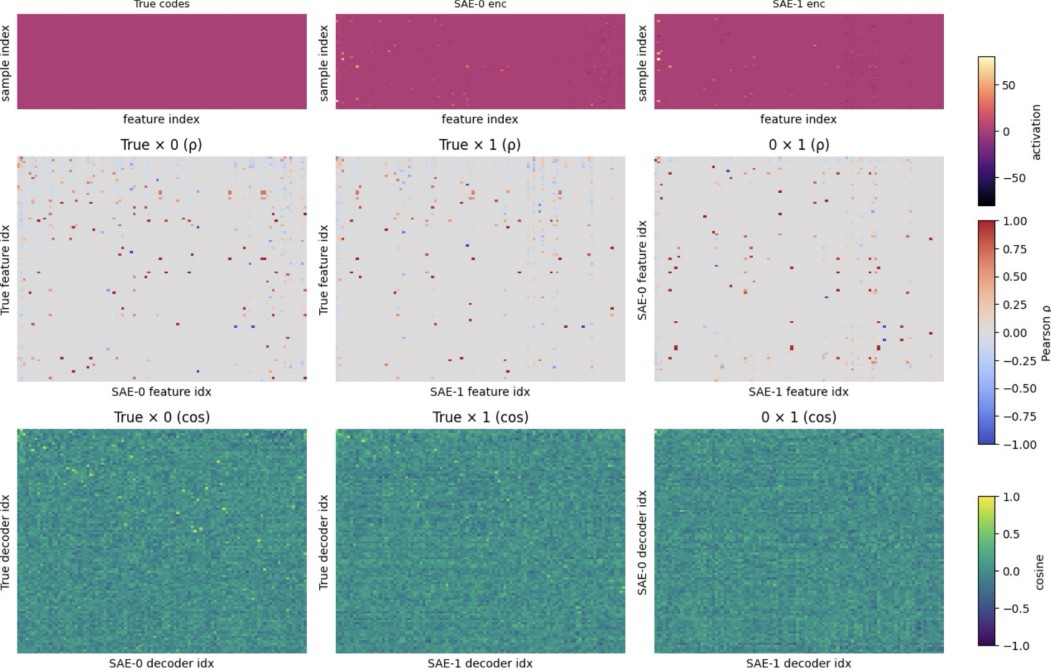

Figure 7: **Random MSAE (example seed pair).** Same layout as Fig. 5; top—activation rasters; middle—activation–Pearson; bottom—decoder–cosine; columns are ($0$ vs. $Y^*$), ($1$ vs. $Y^*$), ($0$ vs. $1$).

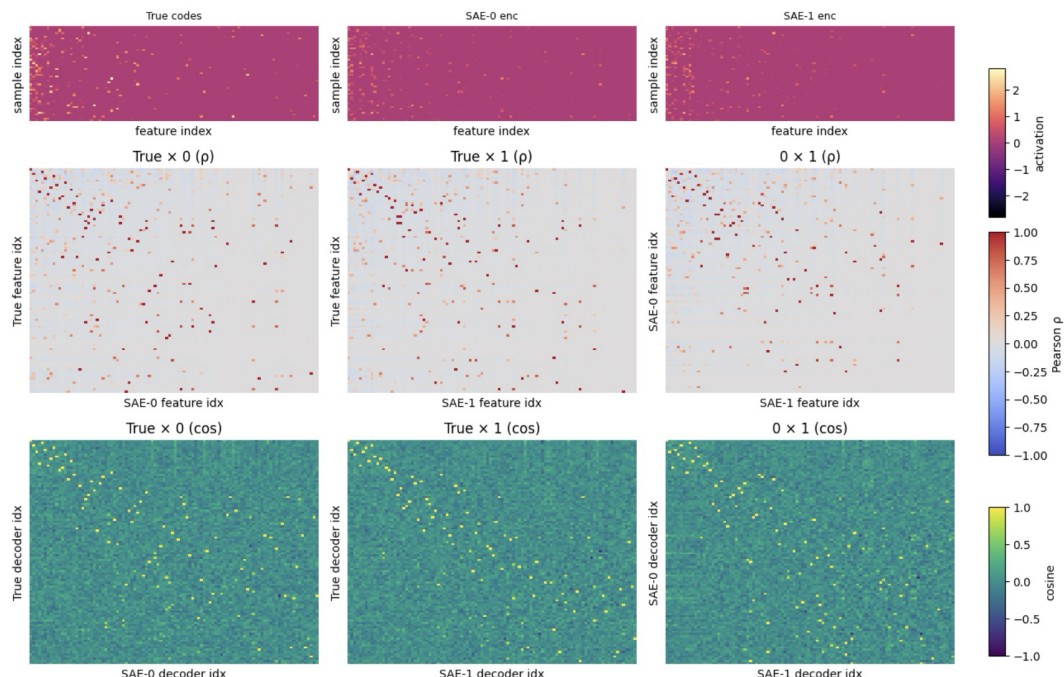

Figure 8: **O-SAE (example seed pair).** Same layout as Fig. 5. Top row rasters (50 inputs); middle row all-pairs activation–Pearson for $\mathrm{Stab}_Z/\mathrm{Ord}_Z$; bottom row all-pairs decoder–cosine extending Fig. 4 to all pairs.

we draw $N = 50{,}000$ samples. Unless noted otherwise: Gaussian features, Zipf exponent $\alpha$ swept across panels, 30,000 training steps, learning rate $10^{-4}$, $\ell_1$ coefficient 0.01, and results are averaged over 5 seeds. We compare to TopK, two Matryoshka variants ("fixed" and "random"), and a vanilla SAE. We weighted the features for Matryoshka and Ordered SAEs by the Zipfian alpha value of the data. For fixed Matryoshka, we used 8 groups. For random Matryoshka, we used 4 truncations.

**Results.** Figure 10 shows that O-SAEs achieve *consistency* comparable to the strongest baselines: for small $\alpha$ (near-uniform usage) ground-truth stability is $\approx 0.9$ for O-SAE, TopK, and Matryoshka, and remains competitive as skew increases. Unlike TopK and vanilla, O-SAEs also exhibit *orderedness*: the learned atom ordering correlates with the ground-truth order (Spearman $\rho \approx 0.5$ at low $\alpha$, remaining positive across the sweep), while pairwise orderedness likewise improves relative to vanilla. This ordering bias comes with a trade-off in global $\ell_2$ reconstruction error, where O-SAEs are higher than TopK/vanilla.

A final observation is that our *Frequency-Invariant Feature Reconstruction (FIFR) Error* (Sec. B.4) tracks dictionary stability across methods and $\alpha$ much better than the global MSE. In ordered/Zipfian regimes, rare features contribute little to MSE and can be underfit without a visible penalty, whereas FIFR Error exposes such failures. Empirically, when TopK and O-SAEs attain a FIFR Error comparable to vanilla SAEs, their ground-truth stability also converges to vanilla, despite differences in global MSE.

Going forward, we hypothesize that with better hyperparameter tuning and methods such as unit sweeping, it is possible to achieve lower L2 and FIFR error with O-SAEs and thus higher orderedness than baseline architectures while maintaining 0.9 consistency.

B.4 MSE UNDERWEIGHTS RARE FEATURES; A FREQUENCY-INVARIANT ERROR

**Motivation.** In ordered/Zipfian settings, some features appear far more often than others. The global reconstruction MSE $\mathbb{E}\|x - \hat{x}\|_2^2$ therefore emphasizes frequent features and can

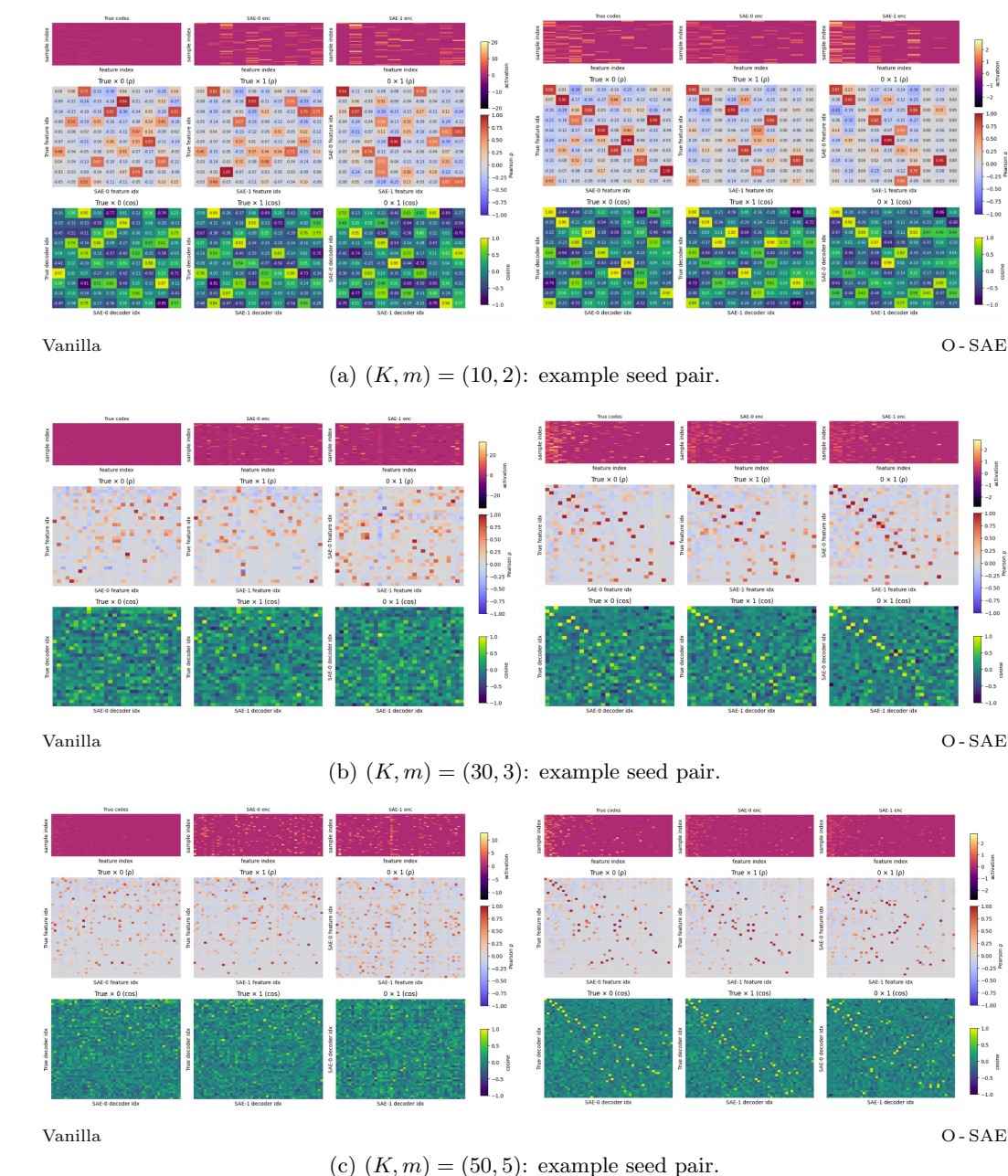

(a) $(K, m) = (10, 2)$: example seed pair.

(b) $(K, m) = (30, 3)$: example seed pair.

(c) $(K, m) = (50, 5)$: example seed pair.

Figure 9: Top: activation rasters for 50 inputs; middle: all-pairs activation–Pearson; bottom: all-pairs decoder–cosine. Columns within each row are $(0 \text{ vs. } Y^*)$, $(1 \text{ vs. } Y^*)$, $(0 \text{ vs. } 1)$.

look "good" while rare features are poorly reconstructed. We seek a metric that (i) treats each feature equally regardless of frequency and (ii) scores the fidelity of its *per-feature* contribution to the reconstruction.

**Definition (Frequency-Invariant Feature Reconstruction *Error*).** Let the ground-truth dictionary be $A^\star \in \mathbb{R}^{m \times n}$ with atoms (rows) $a_j^\star$, true codes $S^\star \in \mathbb{R}^{N \times m}$, learned decoder $A \in \mathbb{R}^{m \times n}$ with atoms $a_k$, and inferred features $F \in \mathbb{R}^{N \times m}$. We align atoms by Hungarian

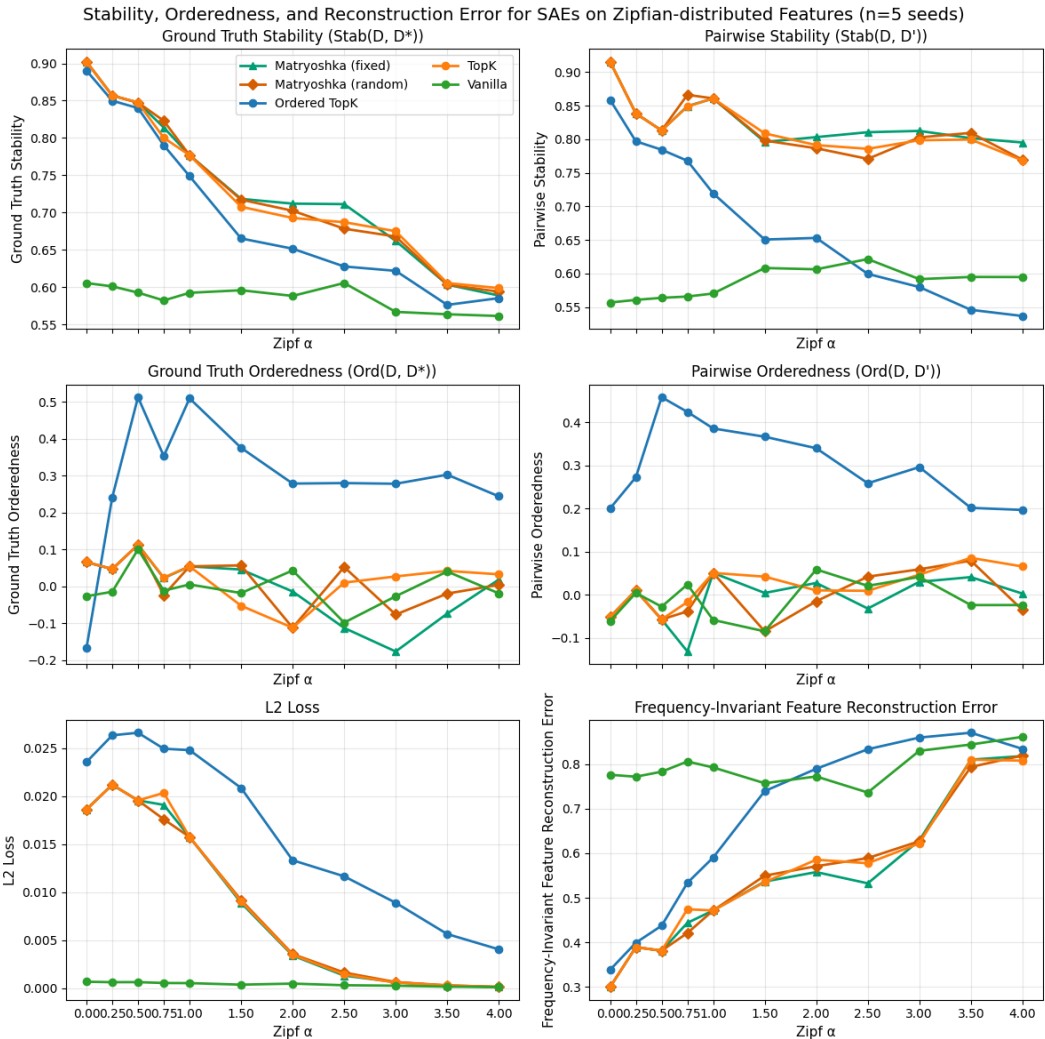

Figure 10: **Zipfian toy-model comparison of SAEs (5 seeds).** Each panel sweeps the Zipf exponent $\alpha$ controlling activation skew (higher $\alpha \Rightarrow$ rarer tail features). *Top row:* O-SAE (Ordered TopK) matches TopK/Matryoshka on ground-truth and pairwise stability ($\sim 0.9$ at low $\alpha$, remaining competitive as skew rises). *Middle row:* O-SAE exhibits positive orderedness (Spearman $\rho$ with ground-truth ordering $\approx 0.5$ at low $\alpha$; pairwise orderedness likewise improves), unlike vanilla/TopK. *Bottom row:* O-SAE has higher global $\ell_2$ error, but the proposed *Frequency-Invariant Feature Reconstruction (FIFR) Error* better predicts stability across methods and $\alpha$: when FIFR Error aligns across methods, ground-truth stability aligns as well. Experimental details: input dim 16, dictionary size 32, $k = 3$, $N = 50$k, Gaussian features, 30k steps, $\text{lr} = 10^{-4}$, $\ell_1 = 0.01$.

assignment on absolute correlations of $\ell_2$-normalized atoms:

$$\tilde{a}_j^\star = \frac{a_j^\star}{\|a_j^\star\|_2}, \quad \tilde{a}_k = \frac{a_k}{\|a_k\|_2}, \quad C_{jk} = \langle \tilde{a}_j^\star, \tilde{a}_k \rangle, \quad \pi \in \arg\max_\sigma \sum_{j=1}^m |C_{j,\sigma(j)}|.$$

For feature $j$, let $I_j = \{i : s_{ij}^\star \neq 0\}$. Define true and estimated per-sample components

$$c_{ij}^\star = s_{ij}^\star a_j^\star, \qquad \hat{c}_{ij} = f_{i,\pi(j)} a_{\pi(j)}.$$

With $\varepsilon = 10^{-12}$,

$$r_j = \frac{\frac{1}{|I_j|} \sum_{i \in I_j} \|c_{ij}^\star - \hat{c}_{ij}\|_2^2}{\frac{1}{|I_j|} \sum_{i \in I_j} \|c_{ij}^\star\|_2^2 + \varepsilon}, \qquad \mathrm{FIFR}(A^\star, S^\star; A, F) = \frac{1}{|J|} \sum_{j \in J} r_j, \ J = \{j : |I_j| > 0\}.$$

**Properties.** (i) *Frequency-invariant:* macro-averaging across features prevents frequent atoms from dominating. (ii) *Per-feature scale-invariant:* normalizing by the energy of $c_{ij}^\star$ removes dictionary–code scaling ambiguity. (iii) *Permutation-invariant:* alignment via $\pi$ factors out atom ordering. (iv) *Interpretable:* FIFR Error equals 0 iff per-feature components are recovered exactly; larger values indicate worse reconstruction (and can exceed 1). As seen in Fig. 10, FIFR Error correlates strongly with dictionary stability in Zipfian regimes, whereas global MSE does not.

# C    Empirical results

Figure 11 and 12 show how orderedness and stability metrics change as SAEs are trained for O-SAE, Random MSAE, and Fixed MSAE on the Pile and Dolma. Orderedness and stability tend to increase over the 45M tokens illustrated in the figures, with some exceptions at lower prefix lengths going down while higher prefix length measures increase.

Figure 11: Prefix $\mathbf{Ord}(D,D')$ plotted with increasing training tokens. Top row is trained on the Pile, and the bottom row is trained on Dolma. (n=1 pair of seeds)

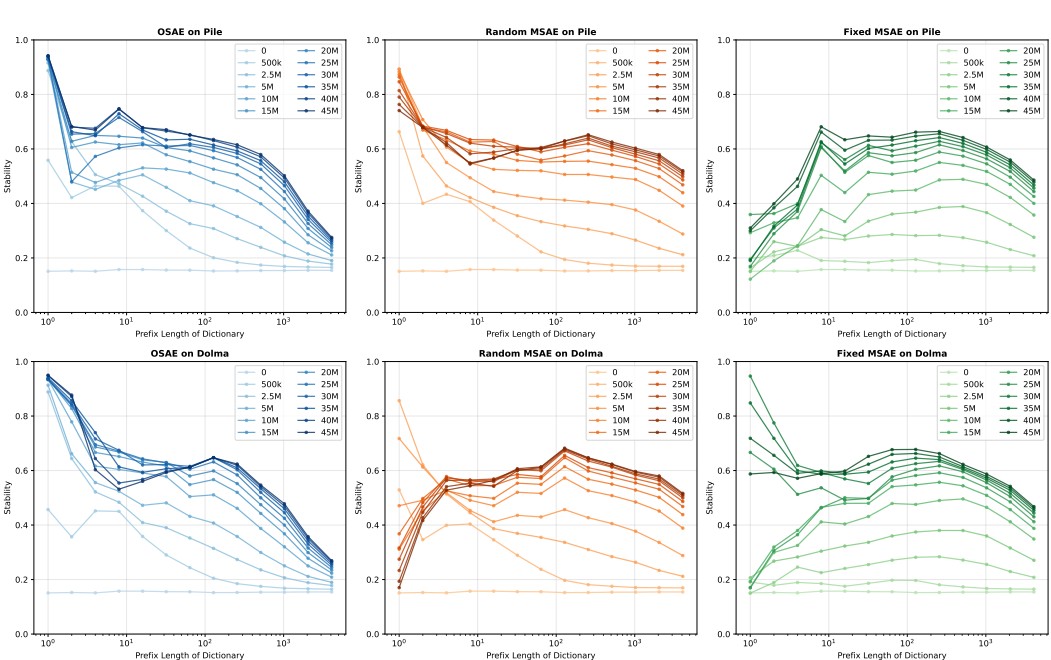

Figure 12: Prefix $\mathbf{Stab}(D, D')$ plotted with increasing training tokens. Top row is trained on the Pile, and the bottom row is trained on Dolma. (n=1 pair of seeds)

## D   Empirical results - SAE Stitching

**O-SAEs decrease the number of novel features found by SAE Stitching**

A key limitation of standard SAEs is their incompleteness, since they often fail to recover the full set of canonical features in a model's representations. Prior work Leask et al. (2025) highlights this issue using SAE stitching. In this procedure, we take a feature (latent) discovered by a larger SAE and "stitch" it into a smaller SAE. If this stitched latent improves reconstruction performance, it suggests that the smaller SAE was missing this information entirely, which means the larger SAE has uncovered a novel feature. If reconstruction worsens instead, the stitched latent is overlapping with existing ones, which means the SAE is redundantly encoding the same information. We call this a reconstruction feature.

Our experiments show that O-SAEs substantially reduce the fraction of novel features discovered via stitching. In other words, O-SAEs capture more of the underlying structure up front, leaving fewer important features uncovered compared to standard SAEs. This reduction in incompleteness directly addresses one of the main critiques of sparse autoencoders: while traditional SAEs leave gaps in the feature set, O-SAEs close those gaps by providing a more complete and less redundant decomposition.

| SAE Type | Novel Feature % | Reconstruction % | No MSE Change % |
|---|---|---|---|
| BatchTopK | 73.8% | 21.2% | 5.0% |
| Random MSAE | 52.7% | 11.4% | 35.9% |
| O-SAE | **33.8%** | **64.8%** | 1.4% |

Table 2: Novel Feature, Reconstruction, and No MSE Change Percentages of various SAE types when stitching 65536-sized features into the corresponding 4096-sized SAE.

In Table 2, the BatchTopK baseline demonstrates 73.8% novel features, indicating strong incompleteness. While O-SAE's 33.8% novel features are still substantial but better than the baseline. Random MSAE falls in between at 52.7% novel features with the caveat that

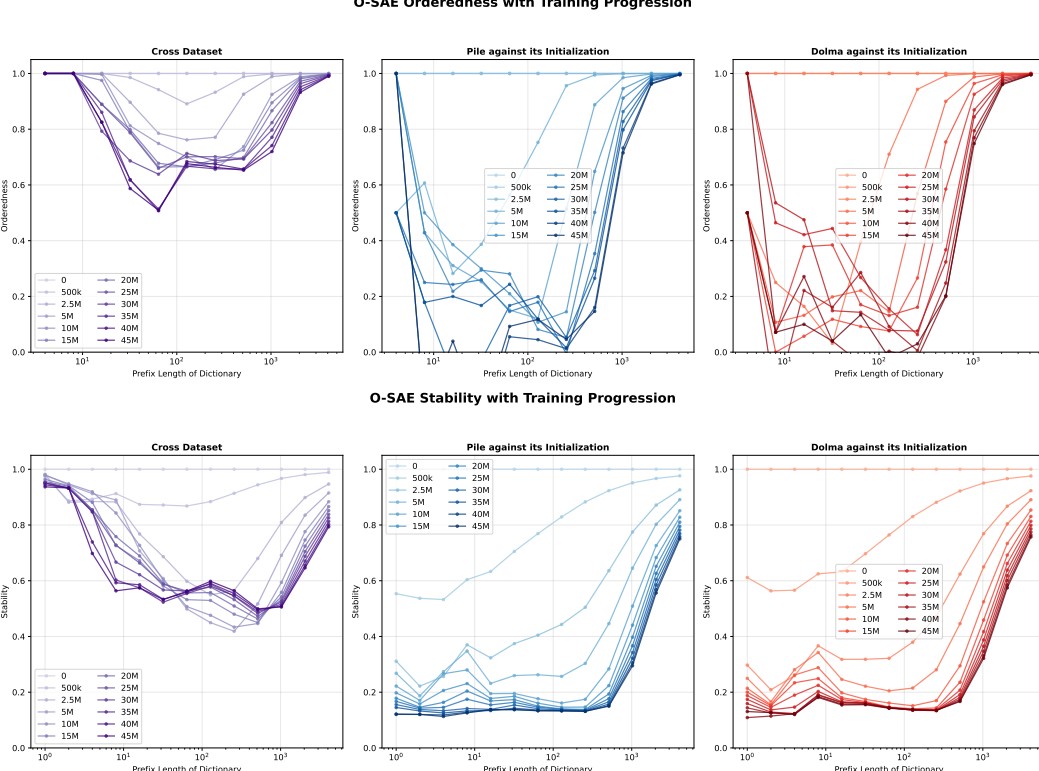

Figure 13: O-SAE Orderedness and Stability. (left) Cross dataset compares O-SAE trained on Pile against O-SAE trained Dolma. They use the same seed, so initial checkpoints start with 1.0 orderedness and stability. (middle) Shows the progression of checkpoints from O-SAE trained on the Pile, when compared against its initialized checkpoint. This gives a relative measure of deviation from initialization. (right) Progression of checkpoints trained on Dolma compared against its initialized checkpoint.

Figure 14: Random MSAE Orderedness and Stability. (left) Cross dataset compares Random MSAE trained on Pile against Random MSAE trained Dolma. They use the same seed, so initial checkpoints start with 1.0 orderedness and stability. (middle) Shows the progression of checkpoints from Random MSAE trained on the Pile, when compared against its initialized checkpoint. This gives a relative measure of deviation from initialization. (right) Progression of checkpoints trained on Dolma compared against its initialized checkpoint.

it has a higher fraction of non-activating features for the limited number of samples tested on. This shows how increasing degrees of hierarchy decrease the novel percentage between 65536 and 4096 sized SAEs.

