# OpenReview forum: "Enforcing Orderedness in SAEs to Improve Feature Consistency"
_ICLR.cc/2026/Conference — Submitted to ICLR 2026_

### Official Review · Reviewer_LFFj · 2025-10-23

**Soundness:** 2
**Presentation:** 2
**Contribution:** 1
**Rating:** 2
**Confidence:** 4

**Summary:**

The authors introduce ordered sparse autoencoders, a variant of Matryoshka sparse autoencoders intended to create greater consistency across seeds and hyperparameter settings for features and their order. They do this by randomly sampling prefixes of the SAE and training based on the reconstruction error of each prefix. This encourages important concepts to appear earlier, without the permutation invariance created by the tiered formulation of Matryoshka.

**Strengths:**

See below

**Weaknesses:**

As far as I can tell, the technique introduced in this paper seems equivalent to the random-prefix variant of Matryoshka SAEs introduced in the Matryoshka SAE paper and used in Section 4.1 on their toy model. They note in the main body that they chose not to use that setup in their experiments in Section 4.2 on the real language model because of worse performance; see Appendix G.4 of that paper for more details.

Leaving aside the novelty, I found the motivation somewhat unclear. Order and stability don't seem intrinsically important to me. A fixed random seed does well on both but hardly feels like an improvement.

I would be sympathetic to an argument that order and stability lead to higher-quality sparse autoencoders, but I would have wanted to see an evaluation on something like SAE Bench on a real model. Toy models are somewhat unreliable, as it is unclear how good a proxy they are for the real thing. Another possible form of improvement would be presenting qualitative evidence of pathologies that arise from non-determinism and disappear with ordered SAEs.

Overall, I am not comfortable recommending acceptance for this paper, as it does not seem particularly novel and does not present sufficient arguments for why the properties introduced by this architecture are desirable.

**Questions:**

Please let me know if I have misunderstood any aspect of the work!

---

### Official Review · Reviewer_yEuK · 2025-10-27

**Soundness:** 2
**Presentation:** 3
**Contribution:** 2
**Rating:** 2
**Confidence:** 2

**Summary:**

The authors note that the solution space of SAE feature reconstruction is generally large and that this can cause poor reproducibility of features across multiple seeds. Matroyshka SAEs go some way to mitigating this by attempting to gain some notion of hierarchy in feature learning. This paper introduces Ordered SAEs (OSAEs) which enforce a strict ordering of latent dimensions via an adaptation of nested dropout (Rippel et al, 2014). The result is gaining more identifiability and better consistency compared to Matryoshka baselines. The authors present theoretical results for ordered unsupervised feature learning via the nested dropout loss. They also demonstrate improvements in feature consistency using their approach on language models. They do not present results in terms of accuracy for language models or any non-toy models however.

Overall, this Is an interesting paper which seeks to solve the problem in SAEs of limited consistency across different random seeds and datasets. The authors approach seems at least partially successful in achieving this aim and explain their methodology effectively. However it is not clear if this method is an improvement over previous SAE architectures on the task of understanding representations in foundation models e.g. LLMs.

**Strengths:**

- Useful recap of related work and framing the problem in context
- I agree that the focus on consistency is a good motivation especially if we are hoping for SAEs to learn a good representation of their target data (e.g. natural language)
- Clear goal with an interesting and well-motivated way to solve the ordering consistency problem
- Mathematically clear and understandable
- Ordering seems like a useful mechanism to encourage identifiability in models and their Orderedness metric seems like a good way to quantitively measure this
- The cross-dataset experiment is useful for understanding what features and not merely facts about a single dataset

**Weaknesses:**

- From Figure 2b, it seems like the method might scale poorly - when the dictionary prefix size is 4096 (a small dictionary size by the standards of the recent SAE literature), the stability in the OSAE case falls dramatically.
    - Though the mathematical results were about the sparse coding case, Figure 2 presents a chart where the dictionary size is only 2x the embedding size of Gemma2-2B (2048) and the stability seems to be good only for <1024 prefix dictionary size
    - This possibly suggests that for stability, this approach is better in the PCA case than the sparse coding case
    - Did you consider a hybrid approach using the O-SAE method for the highly active features (with low prefix length) and the MSAE for the more sparsely active features? It seems like the O-SAE method is quite successful for the most activating (plausibly most important) features
- There are no accuracy comparisons between O-SAEs and other SAE architectures on a language/image modelling task which would be useful to understand the practical utility of their method
    - Hence it’s not clear if the method is a practical improvement over previous SAE architectures for interpretability and disentanglement learning
    - Testing their approach on SAE-Bench or a similar benchmark or providing some Pareto plots could be valuable to show the method’s utility
- Similarly there are no experiments on the interpretability of the learned features either with a human interpretability study or with automated interpretability as in Paulo et al 2025.
- Nit:
    - Some of the citations are improperly formatted (citep vs citet issues) e.g. line 133.
    - On Table 1 it might be worth adding arrows to suggest when a metric is to be maximised vs minimised
- It's not immediately obvious that the high frequency features are the most semantically important, especially in a given domain. For example, for safety applications a somewhat rare event might be the most important to interpret. It would be interesting to hear your thoughts in on whether the orderedness metric is capturing importance in the folk sense of the term
-

**Questions:**

- A strict ordering of latent dimensions provides a very strong inductive bias: do we know when this bias is justified and when enforcing this bias makes the network less flexible and hence more likely to develop solutions with poor reconstructive performance?
- Similarly the conditions from lines 212-213 intuitively seem rather strong, are these conditions approximately true in the case of language models?
- As I understand it the conditions on line 220 is saying that the atoms are ordered by frequency of activation? Is this a correct reading? If so it might be worth clarifying this is the sense of ordering that you have in mind.
- Lines 323-249 state: “We note that O-SAEs have slightly worse reconstruction loss compared to baselines…thus we compare checkpoints with similar loss values for a fair orderedness and stability comparison.”
    - Is the same amount of compute used for the baselines and the OSAE approach?
    - I believe that comparing checkpoints with similar loss values if better loss values are readily available for the baselines could amount to comparing against improperly chosen baselines. Could you provide a comparison against FLOP-matched baselines rather than loss-matched ones?
    - Similarly could you provide the reconstruction loss values for OSAE vs the baselines in the language model representations case? I believe that for most practitioners this is the most important empirical result for your method. Even better would be to include a Pareto curve similar to Gao et al 2024, which details how effective your approach is at many different sparsity values compared to the baseline approaches.
- To better understand the limitations section it would be great to have an estimate for the FLOPs and wall-time cost difference between training Top-K SAEs, MSAEs and O-SAEs. Currently line 428 mentions that there is increased cost but it would be good to quantitively specify this.
    - In particular it would be good to know how the cost scales as for large models with tens or hundreds of thousands of features the asymptotic computational cost could be important.
- Similarly it would be useful to understand how stable/brittle the approach is with hyperparameter variation
- The O-SAE objective optimises for prefix agreement which seems valuable. Is it possible that this approach Goodharts the Orderedness Evaluation though? Comparing on SAE-Bench would be valuable to give a more holistic evaluation
- Figure 3b shows the Orderedness across datasets. For prefix-length of 4 there's a 1.0 orderedness score. Is this to be expected? Similarly for the 4k prefix length it's also ~1.0. Could you show the drift of features from initialisation to the final checkpoint? It seems consistent with the data provided that the early features are frozen quickly and never change from their initialisation and the final features receive the least gradient signal and so are also almost unchanged from initialisation.
    - Falsifiying the "no change from init" hypothesis seems like an important analysis to do for understanding the orderedness here. If this hypothesis is shown to be false then it would be useful to provide a reason for why the very low and high frequency features have the highest orderedness.
- Does O-SAE reduce or increase polysemanticity? We can imagine a setting where making the early prefix features so load-bearing to explain lots of the variance polysemanticity here is higher than it would be in e.g. vanilla SAEs
- Do the results of this paper support a possible approach for SAE research of "using this method we can learn the early features structure and then for different seeds and datasets these will be fixed even if the low frequency features are somewhat more variable"?

---

### Official Review · Reviewer_3cXB · 2025-10-31

**Soundness:** 2
**Presentation:** 2
**Contribution:** 3
**Rating:** 4
**Confidence:** 3

**Summary:**

The paper presents a novel improvement of basic Sparse AutoEncoder (SAE), called Ordered Sparse AutoEncoder (OSAE) that enforces feature orderedness in latent reconstruction, thus eliminating permutation ambiguity present in basic SAE. The authors claim this will mitigate certain limitations of contemporary SAE - differing learned features contingent on random seeds of initialization (reproducibility), improving reproducibility.

The paper first establishes mathematical theory that analytically demonstrates OSAE’s ability to reconstruct latent codes without permutation ambiguity, and then presents two evaluation metrics that quantify consistency and orderedness of the latent features learned, which are leveraged in the following sections to evaluate OSAE experiments on synthetic and real data. Within the empirical results presented, the OSAE does demonstrate higher consistency and orderedness compared to past work to improve SAE such as Matryoshka SAE.

**Strengths:**

The paper exists as a natural continuation of ongoing efforts to improve SAE identifiability and reproducibility through solving permutation ambiguity. While the concept itself is not novel, the paper does contribute important new derivations that resolve permutation ambiguities, which is possibly of significant interest.

The paper presents its main mathematical theorems in succinctly clear manner without ambiguity, it makes a strong argument in theoretically establishing that the OSAE is capable of eliminating permutation ambiguity, under assumed mathematical conditions.

**Weaknesses:**

Lack of elaboration on why eliminating permutation ambiguity is of significant practical interest. Usually, in the identifiability literature (eg starting with linear ICA), identification up to permutation is considered a good solution. It seems like the authors care more about stability, which is more interesting. The Hungarian algorithm can find matches despite permutations. But only if the same features are learned. The theory only extend to the ordering, the authors don't explain why the learned features become unique. Moreover, there is existing literature on this, which they should discuss and relate to: https://arxiv.org/abs/1804.03599, https://openaccess.thecvf.com/content_CVPR_2019/papers/Rolinek_Variational_Autoencoders_Pursue_PCA_Directions_by_Accident_CVPR_2019_paper.pdf, https://proceedings.mlr.press/v202/simon23a.html, https://arxiv.org/abs/2506.06489)

Line 166: Need to clarify whether there are additional constraints required for the distribution $p_{\text{NL}(l)}$ besides it having a full support.

For empirical results, the range of number of atoms, as well as the range of sparsity level tested, is limited, and seem to be way below what a common SAE analysis in LLM would assume.

The results compare only consistency and stability, but make little comments on identifiability of the features learned. They could benefit from additional interpretability evaluation analysis.

Visually, the font sizes of all figures presented are too small.

Missing citation on SAE shortcomings and links to compressed sensing: https://arxiv.org/abs/2411.13117

Generally, the impression is that the theory is solid but the experiments are lazy/sloppy. One would hope for exploration of more settings, SAEs, hyperparameter ablations etc. Moreover, some of the plots show somewhat mixed findings.

**Questions:**

Why is achieving feature consistency across different seeds important? Usually one would want to understand all features that can be learned by an SAE.

While the OSAE achieves better consistency and stability, how do we know that the features learned are the ones we find useful?

Why did you not run SAEs on SAEbench for comparison?

---

### Official Review · Reviewer_aopV · 2025-11-01

**Soundness:** 2
**Presentation:** 2
**Contribution:** 1
**Rating:** 4
**Confidence:** 4

**Summary:**

The paper proposes Ordered Sparse Autoencoders, adapting nested dropout to impose a strict prefix ordering over latent units and claiming this resolves permutation non-identifiability under sparse dictionary learning assumptions. The method is motivated by feature instability across seeds, with a theorem showing exact ordered recovery under a spark condition and a frequency ordering prior. Experiments cover a toy Gaussian model and evaluations on Gemma-2 2B and Pythia-70M, reporting higher orderedness and early-prefix stability relative to Matryoshka SAEs, with some reconstruction tradeoffs. The orderedness metric is Spearman rank after Hungarian matching, and stability uses cosine after Hungarian assignment. Figures 2 and 3 visualize the prefix curves, and Table 1 summarizes toy-model gains.

**Strengths:**

I like the clean formalization of prefix-ordered training objectives and the link back to classical identifiability. The theoretical result under a spark condition is great (although not new) and gives readers a crisp if idealized target. The empirical setup tries to separate pairwise stability from ordering and uses cross-dataset checks on Pythia which is a good instinct. The paper is honest about computational cost and sensitivity of design choices, which I appreciate.

**Weaknesses:**

Even if I find the paper easy to read, there is, in my opinion some strong weaknesses (Major M and minor m) and that I will detail here.

M1. False problem risk. The core promise is to avoid post hoc index permutations, but the evaluation already permits a full Hungarian assignment to score stability and orderedness. For many use cases, doing Hungarian after training is cheap compared to SAE training itself, so the practical benefit of enforcing ordering during training remains under-motivated. Please articulate concrete scenarios where training-time ordering changes downstream utility versus a post hoc matching baseline. The paper itself notes training cost increases when covering many prefixes, which deepens this question. I refer the authors to [1] where I think the discussion on stability and the link with energy is useful.

[1] Interpreting the linear structure of vision-language model embedding spaces, Papadimitriou & al.

M2. Missing post hoc baselines. A strong baseline is to train a standard Top-k SAE and compute a multi-run canonicalization by Hungarian matching across seeds, then reuse that mapping for evaluation and analysis. This would isolate the value of ordered training beyond what a cheap matching step already provides. Right now the method advantage could be a proxy for less noise in late units rather than a fundamental win. The metric definitions in Section 2.7 and 2.8 make this comparison very natural.

M3. Motivation for the ordering prior. The theorem assumes a frequency-based ground-truth ordering. In practice it is unclear why frequency or any fixed prefix order should be desirable. Please discuss risk of misspecified orderings, and whether the gains you see are due to curriculum pressure rather than finding a more “canonical” basis. The Limitations section already hints at over-regularization if the order is wrong; this should be surfaced earlier and tested.

M4. Energy filtering baseline. A simple heuristic is to order or weight features by average activation energy, for example mean absolute Z per unit, and evaluate orderedness and stability on the top-energy prefix. This should be included as a baseline, ideally both post hoc and with a training-time loss weight proportional to average energy. The current paper evaluates orderedness only under the proposed prefix distribution and Matryoshka groupings.

M5. Archetypal SAE comparison and initialization. Since Archetypal SAE directly targets stability and concept structure, a head-to-head on your Gemma and Pythia setups would be informative. At minimum try k-means to initialize the dictionary (and encoder) and better if archetypal-style initializations for the decoder as a stronger baseline. The paper cites Archetypal SAE but does not compare empirically.

M6. Evidence that orderedness helps interpretation. If the promise is better interpretability, please provide at least qualitative slices showing that the top-prefix units correspond more consistently to the same human-named features across seeds than Matryoshka or vanilla. I think you are correct, but the data you currently have do not show that.

Now for the minor concerns,

m1. Tighten the motiviation in the Introduction with concrete downstream tasks where stable indexing matters during training rather than after training. Cite cases where indices must be fixed for intervention or steering.

m2. Please make the Hungarian-based orderedness definition self-contained when introduced. Right now the reader jumps between stability and orderedness and both rely on Hungarian assignment; a small boxed definition could help.

m3. Style. In my opinion,  the paper reads a bit like a laundry list in some places. Consider a tighter narrative arc and move some ablations to the appendix. Again it is well written, but maybe spend a bit more time to create a narrative, for example, a short running example would reduce cognitive load.

m4. Related work. Expand coverage on the link between dictionary learning and SAEs, and position your work more directly relative to your aim of stability and orderedness. The references section shows awareness but the comparison is thin.

**Questions:**

See my Major point 1-6 and minor. 1-4.

---

### Meta-Review · Area_Chair_QV47 · 2026-01-15

**Summary:**

Rejection is recommended. Reviewers (Scores: 4, 4, 2, 2) argued that the core motivation is weak, as cheaper post-hoc matching renders training-time ordering unnecessary. Furthermore, the method lacks significant novelty compared to "random-prefix" Matryoshka SAEs and the evaluation was criticized for relying on toy models without standard benchmarks or interpretability gains.

**Reviewer Concerns:**

As no rebuttal was provided, all major concerns remain outstanding:


Motivation: The necessity of enforcing order during training rather than using post-hoc alignment remains unjustified.

Novelty: The method's similarity to existing Matryoshka variants is unresolved.

Evaluation: The lack of scaling, accuracy metrics, and standard benchmarks (e.g., SAE Bench) persists

**Reviewer Scores:**

Reviewer scores would likely remain unchanged. The critique regarding the "false problem risk" and the insufficient experimental scope  represent fundamental flaws that were not addressed.

---

### Decision · Program_Chairs · 2026-01-26

Reject